



# Modeling nitrous oxide emissions from agricultural soil incubation experiments using CoupModel

Jie Zhang[1], Wenxin Zhang[2], Per-Erik Jansson[3], Søren O. Petersen[1]

[1]Department of Agroecology, iClimate, Aarhus University, Tjele, Denmark
[2]Department of Physical Geography and Ecosystem Science, Lund University, Lund, Sweden
[3]Department of Sustainable Development, Environmental Science and Engineering, KTH Royal Institute of Technology, Stockholm, Sweden

*Correspondence to*: Jie Zhang (jiezh@agro.au.dk)

**Abstract.** Efforts to develop effective climate mitigation strategies for agriculture require methods to estimate nitrous oxide
(N$_2$O) emissions from soil. Process-based biogeochemical models have been used for such estimations but were mainly tested with field-scale measurements. In this study, results from a short-term (43-day) factorial incubation experiment were used to investigate the ability of a process-oriented model (CoupModel) to estimate N$_2$O and carbon fluxes, and soil mineral nitrogen (N) dynamics. This study identified the sensitivities of model parameters when estimating three output variables using a global sensitivity analysis approach. Our results suggested that important parameters regarding N$_2$O flux estimates
were linked to the decomposability of soil organic matter (e.g. organic C pool sizes) and the denitrification process (e.g. Michaelis constant and denitrifier respiratory rates). The model was able to simulate low-magnitude daily and cumulative N$_2$O fluxes with model errors (MEs) close to zero, but tended to underestimate N$_2$O fluxes as observed daily values increased over 0.1 g N m$^{-2}$ day$^{-1}$. Besides, the response of N$_2$O emissions to soil moisture was not well reflected in the model, probably related to the indirect involvement of soil moisture response function in the denitrification process. We also
evaluated ancillary variables regarding N cycling, which indicates that more frequent measurements and additional types of observed data such as soil oxygen content and the microbial sources of emitted N$_2$O are required to further evaluate model performance and biases. The current description of the N cycling process in the model may not consistently represent the temporal scale of nitrification and denitrification processes behind N$_2$O emissions. The major challenges for calibration are associated with high sensitivities of denitrification parameters to initial soil moisture abiotic conditions and residue
amendment. For the development of process-based models, we suggest there is a need to address soil heterogeneity, and to revisit current subroutines of moisture response functions.

## 1 Introduction

The potent greenhouse gas nitrous oxide (N$_2$O) has been estimated to be responsible for about 7 % of the overall global radiative forcing by long-lived greenhouse gases (World Meteorological Organization, 2021). N$_2$O emissions from the
agricultural sector account for 60-70 % of the total anthropogenic emissions of this gas (Davidson and Kanter, 2014; Syakila



and Kroeze, 2011). To provide a scientific basis for developing achievable climate mitigation strategies, improved understanding of $N_2O$ production in agricultural soils and quantification of $N_2O$ emissions are urgently needed.

$N_2O$ emissions from agricultural soils are driven by a suite of microbiological processes among which nitrification and denitrification predominate as sources of $N_2O$. The factors directly regulating nitrification and denitrification activity are the availability of mineral nitrogen (N), oxygen, and degradable carbon (C) sources used by denitrifying organisms (Wijler and Delwiche, 1954). Indirect controls include soil temperature, moisture, pH, and soil texture. During nitrification, where ammonia ($NH_3$, at equilibrium with ammonium $NH_4^+$) is oxidized to nitrate ($NO_3^-$), a small proportion of N may be lost as $N_2O$ (Firestone and Davidson, 1989). Nitrification mainly occurs in well-aerated soils with moderate water content (Goreau et al., 1980; Li et al., 1992; Parton et al., 1996). In contrast, denitrification is a microbial process that occurs under anaerobic conditions where $NO_3^-$ is reduced to gaseous N. Soil C substrates are electron donors for denitrification, but they are also a sink for oxygen that leads to anaerobic microsites (Sommer et al., 2004), and finally $NO_3^-$ is used as electron acceptors. Nitrifying and denitrifying bacteria are most active to produce $N_2O$ in environments with abundant N relative to assimilatory demands by other microorganisms or plants (Firestone and Davidson, 1989), as is often the case following input of fertilizers, manure, or crop residues to the soil.

Farming practices influence the potential for interactions between microbial, physical, and chemical processes in the soil. Incorporation of crop residues can reduce $NH_3$ losses and enhance degradation compared to leaving residues at the soil surface, but the increased soil water holding capacity and oxygen demand locally may stimulate the development of anaerobic microsites and bacterial denitrification activity (Kravchenko et al., 2018; Kuzyakov and Blagodatskaya, 2015). Mechanical disturbance via tillage may influence soil properties (e.g. porosity, aggregate size distribution, solute and gas diffusivities) and microbial enzyme activities, with subsequent changes in the magnitude of $N_2O$ emissions (Grandy and Robertson, 2006).

The quantification of $N_2O$ emissions from agroecosystems is constrained by logistical challenges and resource availability (e.g. analytical equipment and budgets). Process-oriented biogeochemical models, e.g. DNDC (Li et al., 1992), DayCent (Parton et al., 1996), APSIM (Keating et al., 2003), and CoupModel (Jansson and Moon, 2001), have been developed to partly compensate for these limitations. In the application of process-based models, available in-situ measurements can be used to infer model parameters and allow simulation of soil N transformations and $N_2O$ emissions at temporal and spatial scales beyond the monitoring sites, but accurately simulating the magnitude and temporal variability of $N_2O$ fluxes under contrasting contexts still poses a challenge. Those models may provide reasonable estimates of $N_2O$ emissions from soils in a narrow context, usually at specific sites and at annual time scales, but they become less successful at finer time resolution (e.g. diurnal time steps) and at sites different from the pre-calibrated ones. This represents a barrier in evaluating the effects of agricultural land use and management on greenhouse gas emissions (Brilli et al., 2017). Such model errors are often attributed to physical and biogeochemical processes being inadequately represented, which calls for the improvement of



process descriptions beyond parameter optimization (Abdalla et al., 2010; Brilli et al., 2017; Gaillard et al., 2018; Uzoma et al., 2015).

Process models attempt to reproduce the most relevant physical and biogeochemical processes through understanding grounded in the best available theory at the time they were developed, after which some new empirical adjustments were gradually added. CoupModel, used in the current investigation, has a high level of detail on soil physical and abiotic components and has adopted details of submodules of nitrification, denitrification, and gas fluxes from the DNDC model (Li et al., 2000; Norman et al., 2008). The description of $N_2O$ emissions, including the links between soil environmental factors

and biological reactions, is based on a series of hypotheses and results generated from both field measurements and laboratory incubations studies (Li et al., 2000), and the algorithms and parameterization of microbial growth and death dynamics were specifically supported by the latter. While our understanding regarding decomposition and denitrification has advanced in recent decades, the incorporation of state-of-the-art knowledge into process-based models has lagged behind. To test the description of $N_2O$ emissions, it is necessary to apply the model to results from properly controlled laboratory

experiments, where the impact of ill-defined pedo-climatic conditions on model predictability can be minimized (Brilli et al., 2017). This may reveal causal relationships behind gas production and transport in a microcosm representing the ecosystem, and suggest new paths for model development.

The application of process-based models has often been challenged by the paucity of prior information and measurements compared to the model's demands, and this is also the case when applying a model to incubation experiments. One widely-

used model calibration method to bridge the gap between model requirements and available data, and to quantify parameter uncertainties, is "generalized likelihood uncertainty estimation (GLUE)" (Beven and Binley, 1992). During model calibration, uncertainty analysis can help assess whether the model performance is good enough compared to the requirement of the applied use of the model, and to evaluate possible biases in simulations (U.S. Environmental Protection Agency, 2009). This may be facilitated by applying a Global Sensitivity Analysis (GSA), which can rank the sensitivities of

parameters so that the model calibration can focus on the relatively more sensitive parameters (Vezzaro et al., 2012), and thereby the model's uncertainties can be more efficiently constrained. While model processes and performance have been extensively documented, in many studies $N_2O$ emissions alone were used to train and test the subroutines of nitrification and denitrification (Chen et al., 2008). Evaluation under controlled conditions and with ancillary measurements is noticeably lacking, which makes it difficult to identify model structure limitations. Thus, a first step in understanding model

performance may be an evaluation using new datasets that contain different variables linked to N cycling based on targeted laboratory experiments. To our knowledge, no previous study has attempted a systematic sensitivity and uncertainty analysis in the prediction of $N_2O$ emissions based on laboratory incubation results.

For this work, we selected CoupModel which has integrated options for uncertainty estimation and performance evaluation (Jansson, 2012). It has a flexible setting of soil layer thickness down to a scale of mm, which is proper to study soil physical





processes at the scale of incubation experiments. Data sets used in the model were obtained from a 43-day laboratory
incubation using a factorial-based design with various crop residue practices and abiotic factors (Taghizadeh-Toosi et al.,
2021). Specifically, our objectives were (i) to conduct a global sensitivity analysis for parameters in a model setup that can
simulate N cycling under different incubation treatments (ii) to calibrate the model and quantify the uncertainty in the
estimates of $N_2O$ emissions, and (iii) to discuss any model limitations identified and suggest directions for future model

improvement. We hypothesized that the model is able to simulate the daily and cumulative $N_2O$ emissions under contrasting
environments in incubated soil cores. Furthermore, we hypothesized that it would be difficult to constrain the parameters of a
complex model to an unambiguous solution with limited laboratory measurements.

## 2 Materials and methods

### 2.1 Laboratory incubation experiment

In spring 2018, soil used for the experiment was collected from the 0-20 cm tilled layer at the Lönnstorp Field Station,
Sweden. Red beets had been grown in the previous year with no cover crop during winter. The soil is sandy loam (61.8 %
sand, 22.4 % silt, and 15.8 % clay) with a pH of 6.18, C content of 15 g kg$^{-1}$, and N content of 1.49 g kg$^{-1}$. After collection,
the soil was partially dried, stored at –20 °C, and thawed one day before sieving and use for the experiment.

Treatments were prepared with four different soil conditions regarding the moisture level (i.e. 40 or 60 % WFPS) and nitrate

content (i.e. no nitrate addition or addition of $KNO_3$ to 100 mg $NO_3^-$-N kg$^{-1}$ dry wt. soil). Soil cores were prepared by
stepwise packing 1 cm layers of soil to a density of 1.25 g cm$^{-3}$ in cylinders to the height of 8 cm, at each step adding
deionized water or a $KNO_3$ solution. The soil treatments were pre-incubated for one week at 15 °C. The experiment involved
two different crop residues, red clover (RC) and winter wheat (WW). RC residues had a C/N ratio of 17.9, and a moisture
level corresponding to 80 % of the fresh weight. The WW residues had a C/N ratio of 90.9, and the moisture content

corresponded to 20 % of the fresh weight. WW residues had  a higher proportion of lignin and ash (11.7 %) than RC residues
(5.1 %). In the experiment, RC or WW residues were either mixed at a rate of 0.04 g DM cm$^{-2}$ into the soil from 0-4 cm
depth and then repacked, or residues were placed as a layer at 4 cm depth; only results from the mixed treatments were used
in the present study. Incubations with RC and WW took place sequentially, and therefore each residue treatment had its own
set of unamended controls. Thus, in total 16 treatments from the incubation experiment were used for this modeling study,

including unamended soils (as controls) and soil-residue mixtures from either red clover or winter wheat.

All cylinders were covered at both ends with perforated plastic caps and incubated at 15 °C for up to 43 days. Gas sampling
for $N_2O$ and $CO_2$ flux measurements took place ten times, i.e., on day 1, 3, 6, 9, 13, 16, 22, 29, 36, and 43. Gas
concentrations were determined by gas chromatography. Additionally, nitric oxide (NO) fluxes were quantified in four
selected treatments set up separately. Soil mineral N pools in all treatments were measured at four destructive samplings





after 1, 6, 22, and 43 days of incubation. Further details about the experimental treatments, preparations, and analytical

methods are given by Taghizadeh-Toosi et al. (2021).

## 2.2 Model description and simulation setup

### 2.2.1 CoupModel

This study used CoupModel v6.1, which can be downloaded from http://coupmodel.com. A detailed description of

CoupModel can be found in Jansson & Karlberg (2010). The main structure of the model is a one-dimensional vertical soil

profile with user-defined layer thickness and subdivisions. The current setup of CoupModel includes a number of

components, of which the following are linked to $N_2O$ emissions (Fig. 1): (i) soil organic matter (SOM) decomposition and

mineralization; (ii) nitrification and nitrifier growth; (iii) denitrification and denitrifier growth; and (iv) gas diffusion

between soil layers and internal exchange of N trace gases between aerobic and anaerobic micro-sites. In the nitrification

subroutine, CoupModel accounts for response functions of soil temperature, soil moisture, mineral N concentration, and pH.

For denitrification, each step in the chain of denitrification is explicitly calculated, and denitrifier activity is directly

influenced by soil temperature, pH, nitrogen oxides and anaerobic fraction. The anaerobic soil volume fraction is calculated

using the "anaerobic balloon" concept, as implemented in the DNDC model (Li et al., 2000; Norman et al., 2008).

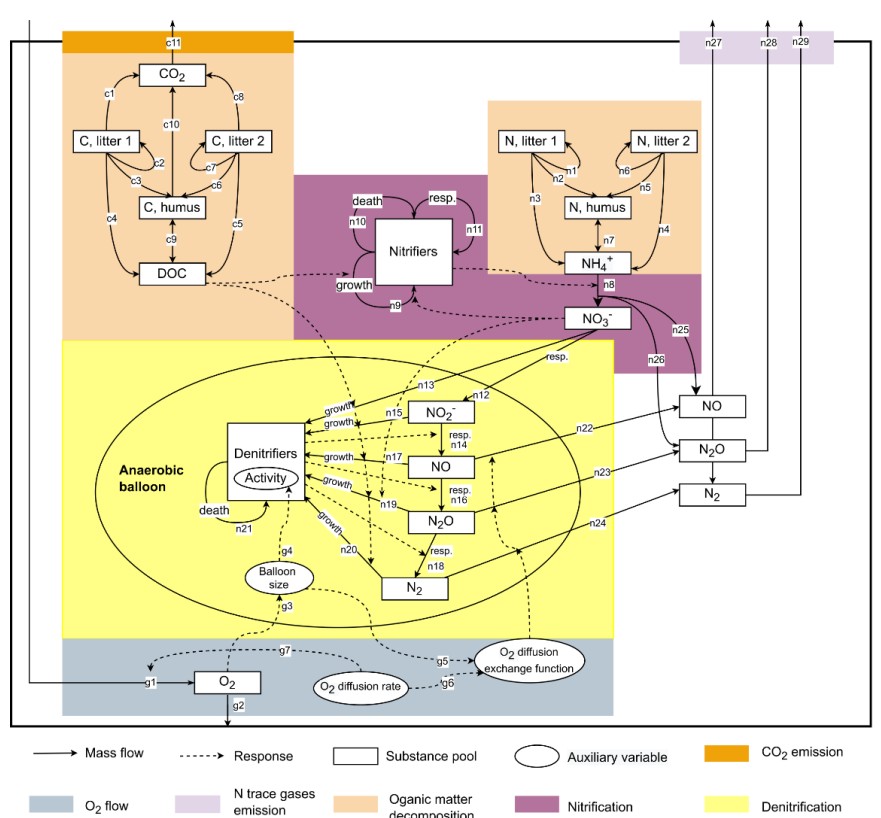



**Figure 1: A conceptual diagram of major C and N processes in the current setup of CoupModel. The details of parameters and equations in each C or N process can be found in Table S2.**

### 2.1.2 Simulation settings

The modeled soil profile consisted of a single soil layer with a depth of 4 cm for the control treatments and 4.2 cm for the mixed treatments, allowing a 2 mm increment owing to residue amendment as observed in the experiment. We only simulated the upper half of the 8 cm soil core since the model only allows external gas exchange at the upper boundary of the soil profile, although in the experiments both ends of the cylinder were exposed to air and the two halves were identical for control soils. For the water process, it was assumed that there was no evaporation from the surface and no vertical water flow across the lower boundary. Constant temperature was set for the upper and lower boundaries, in accordance with incubation conditions. The model was initialized based on the measured soil water content, temperature, pH, total organic C and N, and $NO_3^-$-N, and $NH_4^+$-N of the incubated soil cores. The dynamics of SOM dynamics were simulated with first-order kinetics using three pools (litter1, litter2, and humus). Considering there were no explicit pools designed for crop residue addition, we assigned the rapidly decomposable SOM and metabolic residue materials (e.g. sugars and proteins) to litter1, the moderately decomposable SOM and structural residue materials (e.g. lignin and other fibers) to litter2, and the resistant SOM to humus. For simulating gas transport, we selected the steady-state mode where the oxygen content within the soil profile is a trade-off between soil oxygen consumption and diffusive supply from surface air, and N trace gases are directly lost to the ambient air from the layer in which they are generated.

*Calibration datasets* – Measurements used for model calibration were $N_2O$ flux, $CO_2$ flux, NO flux, $NO_3^-$-N content, and $NH_4^+$-N content. As the gas fluxes and mineral N content in the upper part with soil-residue mixtures and the lower part with bulk soil were not analyzed separately, we assumed that soil C and N turnover in the lower, unamended part was identical to that of control treatments to create datasets for the residue-amended part for modeling. Specifically, the amounts of mineral N and gas fluxes recorded on individual sampling days in the controls were divided by two and subtracted from the values recorded in residue-amended soil.

*Initial values* – (1) Mineral N: Since the mineral N content in the unamended control soil changed little during incubation and the mineral N content in crop residues was negligible, the initial $NH_4^+$-N and $NO_3^-$-N values for the control and residue-amended treatments were taken as the measurement in control soil on day 1. (2) Soil moisture: For the control treatments, the initial volumetric water content was calculated from the water-filled pore space (WFPS) levels of 40 or 60 % according to the total porosity of 0.53. For the residue treatments, the initial volumetric water content was calculated from the moisture content of soil and crop residues (Taghizadeh-Toosi et al., 2021). (3) Organic matter pools: The partitioning of soil organic C between litter1, litter2, and humus was defined by the ratio 0.02:0.54:0.44 (Gijsman et al., 2002). For crop residues, the metabolic fraction of organic C was calculated from the lignin/N ratio: Metabolic fraction = 0.85 - 0.013 (lignin/N) (Gijsman



et al., 2002), and hence the organic C allocation between litter1 and litter2 had a ratio of 0.82:0.18 for RC and 0.55:0.45 for WW. The allocation of organic N in different pools followed the pattern of C and the C/N ratios (Table S5).

A summary of calibration data sets can be found in Table S1 in the supplement, in which cumulative gas emissions were estimated by linearly interpolating between sampling dates and integrating the area under emission curves; and average
mineral N were calculated by dividing the integrated values by the sampling period. The results for nitrate in soil cores with residues were not included due to high uncertainty in the calculations that was probably caused by solute transport between the unamended and amended soil layers, as observed in a related incubation experiment using some of the same soil and residue treatments conducted by Lashermes et al. (2021).

### 2.3 Model sensitivity and uncertainty analysis

**2.3.1 Global sensitivity analysis**

Given uncertain prior information, the study used Morris screening (Morris, 1991) for a global sensitivity analysis to identify the most important input parameters and process parameters affecting $N_2O$ fluxes. We included seven input parameters related to the characteristics of soil and crop residues (i.e. soil porosity, residue porosity, soil pH, and sizes of organic C pools), with realistic ranges of uncertainty intervals considered. Besides, we considered 45 process parameters involved in
the relevant model processes. These parameters are listed in Table S4.

The Morris screening method is a commonly used sensitivity analysis technique, based on an efficient sampling strategy for performing a randomized calculation of one-factor-at-a-time (OAT) sensitivity analysis. This method can be viewed as a compromise between a simple OAT approach and the more complex GSA methods (e.g. variance-based approaches) as it provides a good approximation to the global sensitivity measure of the parameters at an affordable computational cost.
Furthermore, it was considered excessive and unnecessary in the present study to adopt a more detailed analysis given the limited data availability.

The elementary effect (EE) was estimated by comparing the variation of the model's output $y^j$, with the variation of a given parameter $\theta_i$, according to Eq. (1). The number of iterations $n$ was set to 50, and the optimal perturbation factor $\Delta$ was set to 2/3 by dividing the input space into four levels (Morris, 1991). To allow comparison across outputs, the EE was then
standardized by using the standard deviation of the model factor and the standard deviation of the output (SEE, Eq. (2)). The significance of the impact of parameters was tested by comparing the mean of the SEE of those parameters to twice the standard error (*sem*, Eq. (3)) (Sin et al., 2009). If the input factor lies outside this range, it is said to have a significant effect on the output. The codes used in the analysis were adapted from (Sin et al., 2009).



$$EE_i^j = \frac{y(\theta_1^j, \theta_2^j, \ldots, \theta_i^j + \Delta_{OPT}, \ldots, \theta_{m-1}^j, \theta_m^j) - y(\theta_i^j)}{\Delta} \tag{1}$$

$$SEE_i^j = EE_i^j \cdot \frac{\sigma_{\theta_i}}{\sigma_y} \tag{2}$$

$$sem_i = \pm \frac{\sigma(SEE_i^j)}{\sqrt{n}} \tag{3}$$

The GSA was performed to the model-evaluation measure root mean square error (RMSE) for three variables: $N_2O$ flux,

$CO_2$ flux, and soil $NH_4^+$ which had relatively complete measurement data sets. By applying the sensitivity analysis to the
likelihood measure, the main factors that drive the model runs with a good fit to data could be identified (Ratto et al., 2001).
The results from sensitivity analyses were further used to identify process parameters for inclusion in the uncertainty
analysis due to their contribution to output variability.

**2.3.2 Uncertainty analysis**

Model calibration was conducted separately for each of the 16 treatments to give more flexibility in model parameterization.
The calibration was carried out with reference to five measurement variables, namely $N_2O$ flux, $CO_2$ flux, $NH_4^+$ content,
$NO_3^-$ content, and NO flux (only four treatments), using the "generalized likelihood uncertainty estimation" (GLUE)
technique (Beven and Binley, 1992). The GLUE method does not seek the single best fit to the measured data but utilizes an
ensemble of model simulations that represent equally good results using informal likelihood measures, often mentioned as

acceptance criteria. In this study, we first described the entire ensemble of model runs as prior runs and after applying
selection criteria the selected ensemble of model runs was analyzed as posterior runs or behavioral runs. Based on the
calculated sensitivity indices from Morris screening, a total of 26 process parameters were selected for calibration where the
parameters with marginal SEEs were omitted and only one denitrifier growth parameter was kept in each step of the
denitrification chain (see Table S4). These parameters were uniformly or log-uniformly distributed within the predefined

ranges, from which 20,000 parameter sets were then randomly sampled for model runs. Out of these runs, whether a
parameter set was accepted or not was based on the defined criteria, which in this study consisted of coefficient of
determination, $R^2$ and mean error, ME. The latter is defined as ME = E ($O_i$ - $S_i$), where $S_i$ and $O_i$ are the time series of the
simulated and observed data.

The ME acceptance threshold of each variable was set to be around the average of daily measurements taking into account

the different magnitudes of each variable (see Table S3). In prior runs ME values were often skewed to one side (above 0 or
below 0) while setting the same threshold on both sides rejected most of the prior model runs, and hence the ME criterion on
one side might be looser than the other. For $N_2O$ emissions with marked peak fluxes, a combination of $R^2$ and ME was used
for the selection of posterior parameters to simulate the dynamics and magnitudes. An ensemble of ca. 50 posterior runs was




selected with an acceptance rate of 0.25% based on prior simulations. The uncertainties of model predictions were quantified

within the limits and posterior probability distributions of parameters.

Finally, to investigate whether the treatment effects concerning soil moisture and nitrate level could be represented by a common parameterization, we attempted to calibrate process parameters against combined data sets from multiple treatments where measurements from every four treatments with the same residue application were pooled. The prior parameter ensembles used the same 20,000 parameter sets as the single-treatment calibration. Accordingly, the measurement datasets

from the four treatments in each group were pooled and thus a larger data set for model evaluation was obtained. The procedure of selecting behavioral runs followed the aforementioned approach based on ME and $R^2$. A diagram that describes the analysis workflow for this study is presented in Fig. 2.

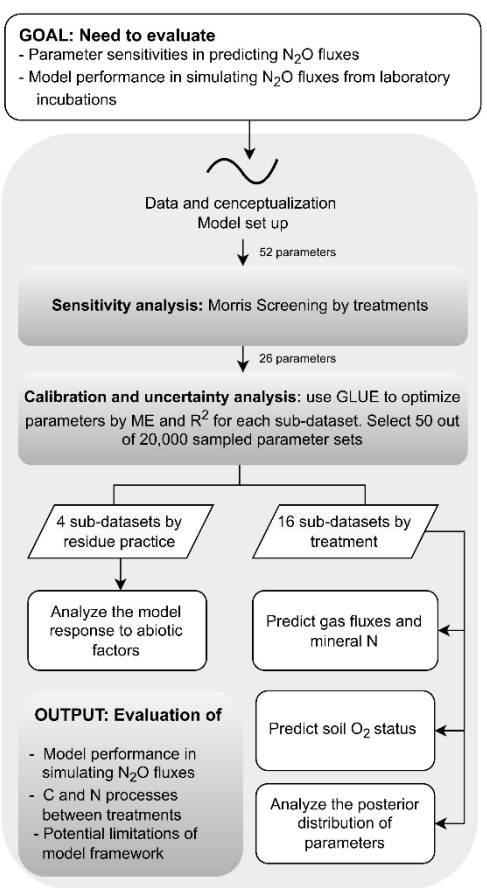

**Figure 2: A schematic diagram for performing sensitivity and uncertainty analyses.**

**3 Results**





## 3.1 Sensitivity analysis

The results of Morris screening were evaluated by comparing the absolute SEE concerning $N_2O$ flux, $CO_2$ flux, and soil $NH_4^+$ for individual treatments. Figure 3a-c lists all parameters ranking in the top five across the 16 treatments. Parameter ranking was performed based on the absolute mean of SEEs – the higher the absolute value, the more important the

parameter is, as shown by the shade of color in Fig. 3a-c. In general, the parameters identified as most influential for soil respiration ($CO_2$ flux) and $NH_4^+$ content showed robustness across treatments as they differed only slightly in their ranking with, respectively, eight and nine different parameters represented. In contrast, more inter-treatment variation was found in the parameter ranking for $N_2O$ emissions with 18 different parameters represented. For $N_2O$, the parameters exhibiting relatively high SEE values for most treatments belonged to categories of SOM decomposition and denitrification (Table S2),

including $d_{effNO}$, $SOC_h$, $d_{growthNO3}$, $cn_m$, and $d_{hrateNxOy}$. The model input, $SOC_h$, characterizing the partitioning of SOM pools in the simulation, was found to be crucial in 13 out of 16 treatments. $d_{effNO}$ represents the respiration of denitrifying bacteria based on NO, and it showed relatively large elementary effects for almost all treatments by directly regulating the reduction step from NO to $N_2O$. The parameter $d_{growthNO3}$ describes the loss of $NO_3^-$ from the anaerobic nitrogen pool due to microbial growth. $d_{hrateNxOy}$ represents the N concentration for half rate in the denitrification process and is also known as the Michaelis

constant of the enzyme (see n13, n15, n17, n19, and n20 in Fig. 1).

Parameters that had the greatest impact on $CO_2$ emissions were concentrated in the following: $SOC_h$, SOM decomposition rates ($k_{l2}$, $k_{l1}$,) and the corresponding efficiencies ($f_{e,l2}$, $f_{e,l1}$). In addition, the two parameters, $p_{\theta Low}$ and $\theta_{wilt}$, controlling the lower limit of the soil moisture response function for the decomposition of organic matter (see Eq. (5.86) in Table S2), exhibited distinct influences for the treatments at the lower moisture level.

The main processes influencing the $NH_4^+$ content of the soil were identified as SOM decomposition and denitrification, and influential parameters included: $cn_m$, $SOC_h$, $k_{l2}$, and $f_{e,l2}$. The C/N ratio of microbes, $cn_m$, has an influence on the mineral N content by changing the magnitude and direction of soil mineralization/immobilization of nitrogen (see n1-n7 and n11 in Fig. 1). It was also found that soil porosity ($\theta_s$) had significant effects on some treatments, especially under higher moisture conditions. Besides, as a key intermediate of mineral nitrogen turnover, the content of $NH_4^+$, was also influenced by

denitrification-related parameters, such as $d_{growthNO3}$.

The average elementary effects across all treatments are shown in Fig. 3d-f. For the $N_2O$ flux, all parameters were located inside the wedge indicating that none of these parameters showed a significant effect across all treatments despite their significance in individual treatments. In contrast, we found that the other two variables, $CO_2$ flux, and $NH_4^+$ content, were significantly affected by 15 and 28 parameters respectively. Moreover, all parameters showed non-linear effects on the

outputs as revealed by their non-zero standard deviations, which suggested that simulated C and N processes did not solely depend on individual parameters but also their interactions.





**Figure 3: Sensitivity analyses for N₂O flux, CO₂ flux, and soil NH₄⁺ content in all treatments. (a-c): Heatmaps that include all parameters ranked in the top five places for each treatment based on absolute standardized elementary effects (SEEs). (d-f): Estimated mean and standard deviation of SEEs averaged across the 16 treatments, where the two lines drawn in each subplot**



**correspond to twice the standard error (sem): $\mu_i = \pm 2sem_i$ (see Sect. 2.3.1): if a factor is located inside the wedge, it indicates that its impact on the output is insignificant and vice versa.**

## 3.2 Uncertainty analysis

### 3.2.1 Temporal dynamics of $N_2O$ flux, $CO_2$ flux, and mineral N

In the experimental treatments with RC amendment, $N_2O$ emission rates were consistently low at 40 % WFPS but were markedly higher and peaked on day 3 at 60 % WFPS (Fig. 5a). The highest measured daily $N_2O$ flux was 1.4 g N m$^{-2}$ day$^{-1}$ in the RC treatment with $NO_3^-$ addition at 60 % WFPS. Similar patterns were observed for $CO_2$ emission rates, with emission peaks at an early stage of incubation (day 1 or day 3) and then followed by a decline. In treatments amended with WW, $N_2O$ evolution rates were generally low compared to those with RC amendment, and showed higher rates at 60 % WFPS and in

$NO_3^-$ amended soil but treatment effects were generally minor. For $CO_2$ evolution, higher rates were detected by day 1, but there was also a secondary peak after 1-2 weeks. The control treatments of the WW residue incubations showed less $CO_2$ and $N_2O$ release compared to the control treatments of the preceding RC incubations.

The prior models generally showed significantly biased mean errors in terms of gas emissions and soil mineral N, and their magnitude was reduced in the posterior models for most model outputs (Fig. 4a-d). For $N_2O$ fluxes, most treatments

amended with WW and corresponding controls did not show significant deviations from the observed fluxes. In contrast, in treatments amended with RC and corresponding controls, though the absolute MEs had been reduced, there were still significant deviations generally in the direction of underestimating the observed fluxes. For $CO_2$ emissions, 13 out of 16 treatments showed reduced mean errors in the behavioral models and half of the treatments showed insignificant deviations from the observed fluxes. For soil $NH_4^+$ content, there was a severe overestimation for most prior models, but this was

alleviated by posterior models, and seven treatments showed insignificant deviations from 0 after model calibration. For soil $NO_3^-$ content in control treatments, the ME ranges of posterior runs were around zero while negative or positive biases existed especially the former. The simulated evolution of associated variables is depicted in Fig. 5a-d and results are summarized below.

*$N_2O$* − The accepted simulations (Fig. 5a) were able to represent the scenarios with low daily $N_2O$ emissions ($10^{-5}$-$10^{-2}$ g N

m$^{-2}$ day$^{-1}$), while simulations failed to capture the large emission peaks (e.g. 1.4 g N m$^{-2}$ day$^{-1}$ and 0.13 g N m$^{-2}$ day$^{-1}$ for the two RC treatments with $NO_3^-$ amendment), or the emission dynamics were reasonably simulated (e.g. $R^2 > 0.4$, see Table S3) but the peak values were lower than observed. The $N_2O$ fluxes obtained from the model tended to increase over time and generally agreed with the observed fluxes in the second part of the experiment.

*$CO_2$* − Overall, the behavioral models mimicked the measured dynamics and magnitude of $CO_2$ emissions well (Fig. 5b).

There were overestimations or underestimations by the model, most pronounced in the early stage of incubation. By day 14, a second peak of respiration was observed for the WW treatments that was not simulated by posterior models.



$NH_4^+$ − For RC residue treatments, net N mineralization was observed from the early to mid-stage of the incubation period, followed by a declining trend, whereas in the posterior models in three of four treatments the simulated $NH_4^+$ predicted a trend of net N mineralization throughout the incubation (Fig. 5c). Such a continuously increasing trend also existed in the

prior runs, which would not be radically altered in the behavioral models by setting a stricter selection criterion for $R^2$. For WW residue and control treatments the measured $NH_4^+$ content was at the detection limit, and the magnitude of the simulated $NH_4^+$ content was either in line with the measurements, or a bit higher.

$NO_3^-$ − The change of simulated daily $NO_3^-$ content generally showed a declining trend for all control treatments, with modeled values comparable to observed data. While in most of the control treatments except for the ones with high moisture

and $NO_3^-$ amendment, the observed $NO_3^-$ levels remained stable or slightly increased during incubation (Fig. 5d). For the treatments amended with crop residues, the simulated $NO_3^-$ content showed a more clear downward trend throughout the period, consistent with $NO_3^-$ being utilized as a substrate of denitrification in the simulation. Though no explicit measurement of $NO_3^-$ within the residue-amended layer in the present experiment, the average $NO_3^-$ within the entire soil core in RC treatments showed net consumption followed by a rebound (Taghizadeh-Toosi et al., 2021), consistent with

observations for the RC-amended layer in a comparable incubation experiment by Lashermes et al. (2021). In view of this, the simulated $NO_3^-$ content in residue treatments exhibiting a continuous decline was probably lower than the actual values.

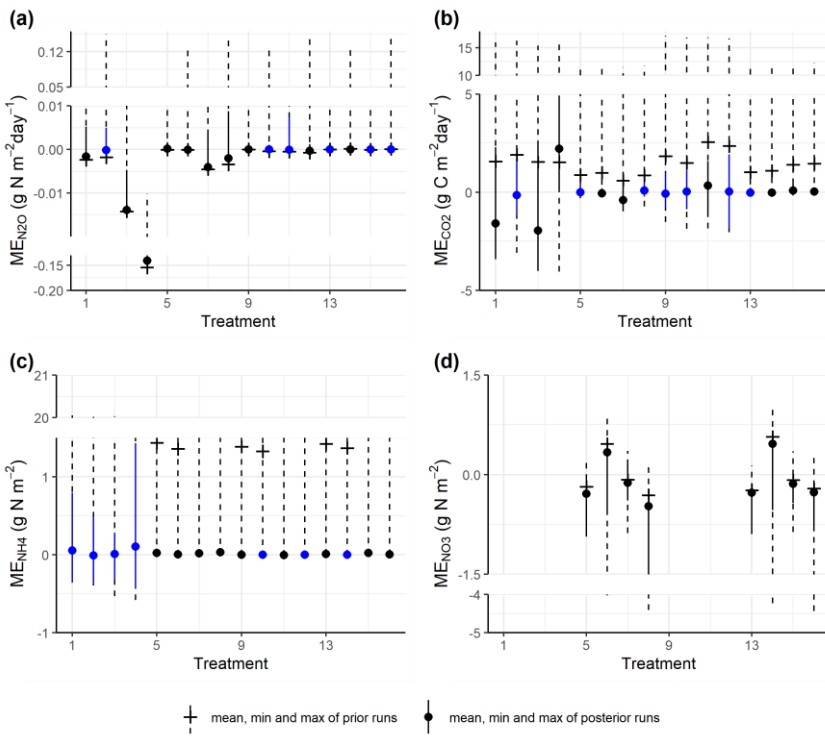



**Figure 4: Comparison of the ME ranges between prior simulations and accepted simulations for daily N$_2$O fluxes (a), CO$_2$ fluxes**
**(b), soil NH$_4^+$ content (c), and soil NO$_3^-$ content (d). Blue color shows ME values not significantly different from zero by one-sample**
**$t$-tests (significance level $\alpha = 0.05$). Treatment indices 1-4 represent treatments of mix RC, 5-8 for control RC, 9-12 for mix WW,**
**and 13-16 for control WW, where treatment conditions are, in order: "40% WFPS, -NO$_3^-$", "40% WFPS, +NO$_3^-$", "60% WFPS, -**
**NO$_3^-$", and "60% WFPS, +NO$_3^-$". No measured data for nitrate in residue treatments.**

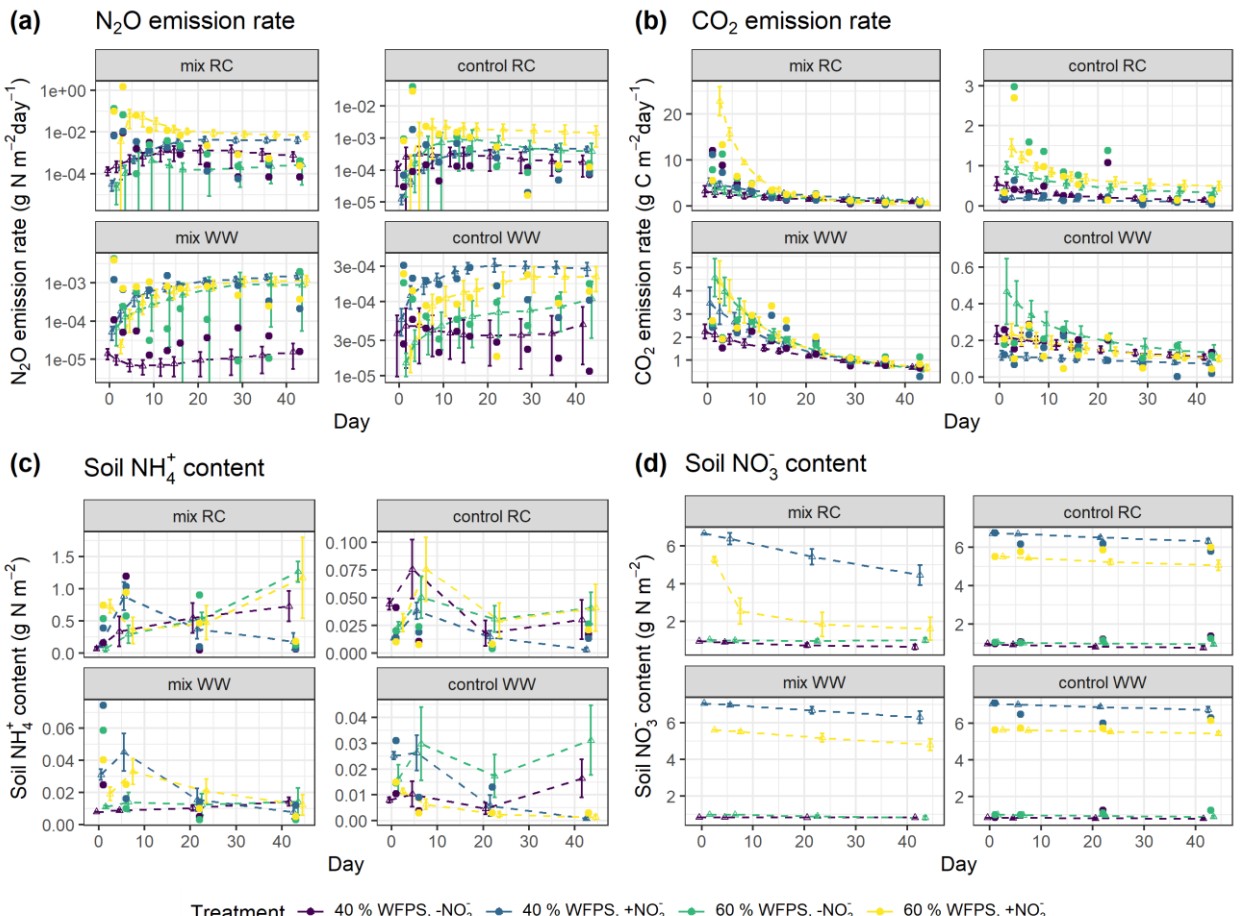

**Figure 5: Simulated and measured daily N$_2$O fluxes (a), CO$_2$ fluxes (b), soil NH$_4^+$ content (c), and soil NO$_3^-$ content (d) during the**
**43-day incubation. Scatter points represent measured data; and triangles with dashed lines represent simulated data (error bar:**
**95 % confidence interval). Daily measurements presented were re-calculated from the data provided by Taghizadeh-Toosi et al.**
**(2021).**




### 3.2.2 Cumulative gas fluxes and average mineral N content

Model predictions of cumulative $N_2O$ fluxes and $CO_2$ fluxes for the 16 treatments were significant and strongly correlated with the observed fluxes (Fig. 6 and Table 1). For $N_2O$, there was a bias towards underestimation of high cumulative $N_2O$ fluxes (slope bias $\beta_1 = 0.17$, ME = -0.23). Estimate of slope in linear regression for cumulative $CO_2$ flux approached $\beta_1 = 1$ indicating there was no consistent bias. For the average $NH_4^+$ and $NO_3^-$ content, the estimated slopes were close to unity, and the deviations between prediction and measurement, signified by the relative RMSEs (rRMSEs), were 46 and 11 %,

respectively. For the average $NH_4^+$ content in the low range, simulated values were found to overestimate the measured data (Fig. 6c).

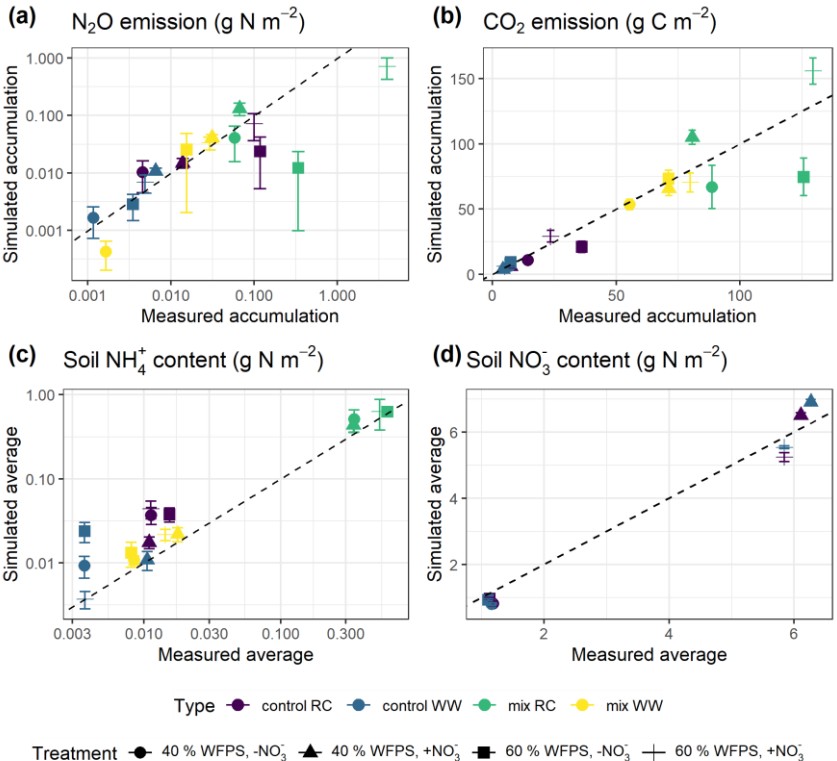

**Figure 6: Simulated and measured cumulative $N_2O$ fluxes (a), $CO_2$ fluxes (b), average $NH_4^+$ content (c), and average soil $NO_3^-$ content (d) during the 43-day incubation (error bar: 95 % confidence interval). Reference lines with a slope of 1.0 are shown on the**

**graphs.**

Table 1: Model evaluation of cumulative gas fluxes and average mineral N content from single-treatment calibration procedure and multi-treatment calibration procedure. The mean values of posterior models were compared with the observed data for 16 treatments. Units are valid for the statistics of ME and RMSE.





| | Cumulative $N_2O$ emission (g N m$^{-2}$) | | Cumulative $CO_2$ emission (g C m$^{-2}$) | | Average $NH_4^+$ content (g N m$^{-2}$) | | Average $NO_3^-$ content (g N m$^{-2}$) | |
|---|---|---|---|---|---|---|---|---|
| Calibration | Single-treatment | Multi-treatment | Single-treatment | Multi-treatment | Single-treatment | Multi-treatment | Single-treatment | Multi-treatment |
| ME | -0.23 | 0.10 | -3.07 | 4.66 | 0.03 | 0.12 | -0.11 | -1.10 |
| RMSE | 0.82 | 0.71 | 17.2 | 12.1 | 0.06 | 0.50 | 0.41 | 0.50 |
| rRMSE | 275 % | 238 % | 34 % | 24 % | 46 % | 411 % | 11 % | 14 % |
| Slope, $\beta_1$ | 0.17 | 0.56 | 0.93[a] | 0.97[a] | 1.15 | 2.13[a] | 1.07[a] | 1.08[a] |
| Intercept, $\beta_0$ | 0.02 | 0.23 | 0.64 | 6.16 | 0.01 | -0.01 | -0.36 | -0.53 |
| $R^2$ | 0.96 | 0.47 | 0.84 | 0.93 | 0.98 | 0.50 | 0.98 | 0.98 |

[a] Values are not significantly different from one by one-sample $t$-tests (significance level $\alpha = 0.05$).

A regression of simulated cumulative $N_2O$ flux residuals against observed data confirmed that underestimations were strongly ($R^2 = 0.92$) associated with the magnitude of observed $N_2O$ fluxes (Fig. S1a). The negative slope of the regression indicated an underestimate of 0.83 g $N_2O$-N for every 1 g of observed $N_2O$-N per square meter. A regression of simulated cumulative $N_2O$ flux residual against the residuals of other variables revealed that underestimations were not strongly associated with the residuals of simulated $NH_4^+$ and $NO_3^-$ (Fig. S1c and d). However, we observed that clustering of residuals concerning mineral N existed in which underestimations of cumulative $N_2O$ flux tended to occur when soil $NH_4^+$ was overestimated and when soil $NO_3^-$ was underestimated. Specifically, residuals for cumulative $N_2O$ flux and soil $NO_3^-$ were simultaneously underestimated in 53 % of the posterior runs as revealed by scatter points falling in the third quadrant; and underestimations of $N_2O$ flux were accompanied by overestimations of soil $NH_4^+$ in 41 % of the posterior runs by looking at scattering points in the fourth quadrant. When only the subset of control treatments was analyzed, the clustering patterns became even more apparent (Fig. S2c and d).

### 3.2.3 Calibration by multiple treatments

Increasing the number of calibration treatments led to reduced uncertainty but, meanwhile, poor performances of posterior models for some treatments (Fig. 7). Cumulative $N_2O$ fluxes were better simulated for the treatments with higher observed fluxes in each group, especially treatments at 60 % WFPS, but were overestimated for others with low observed fluxes. The regression between the mean simulated and measured $N_2O$ flux only accounted for 47 % of the variation in the data, much lower than the level of 96 % in the single-treatment calibration procedure (Table 1). Simulated $CO_2$ and $NO_3^-$ were generally close to the observed data. In the same group, simulated $CO_2$ fluxes were not different between two levels of $NO_3^-$ input but depended on the level of moisture. Simulated soil $NH_4^+$ showed a good agreement with the measured data in the low range of $NH_4^+$ content, but had large model deviations for the four RC residue treatments, in which the $R^2$ was 0.50 in contrast to 0.98 in the single-treatment calibration procedure (Table 1).





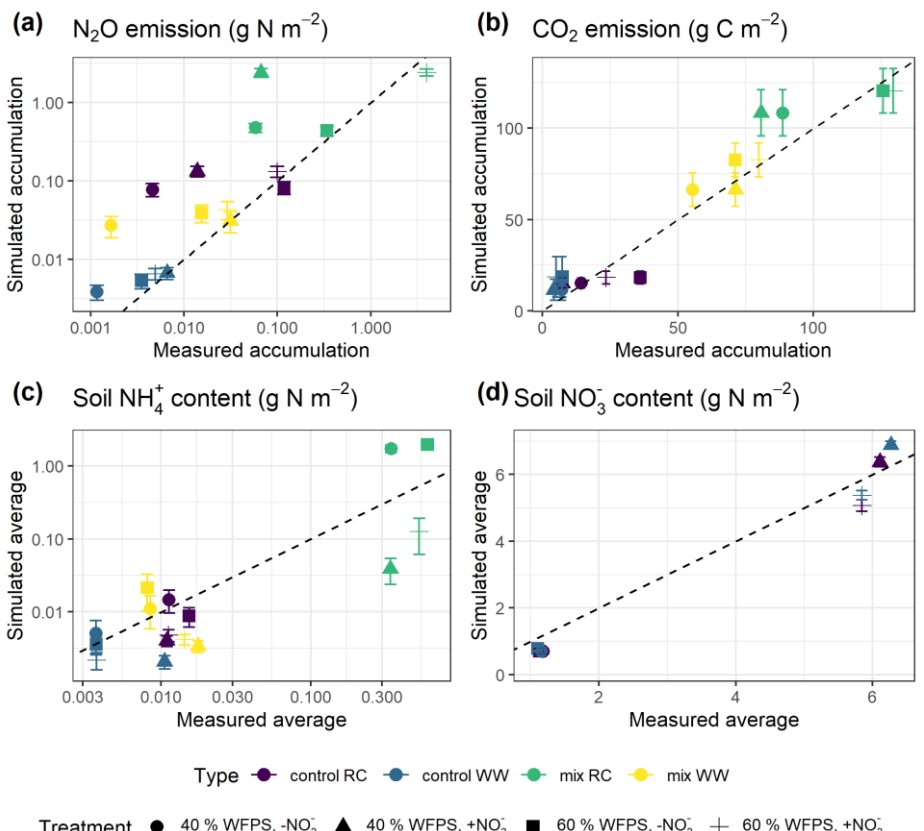

**Figure 7: Simulated and measured cumulative N₂O fluxes (a), CO₂ fluxes (b), average NH₄⁺ content (c), and average soil NO₃⁻ content (d) during the 43-day incubation (error bar: 95 % confidence interval). Simulated results were obtained from multi-treatment calibration. Reference lines with a slope of 1.0 are shown on the graphs.**

### 3.2.4 Simulated oxygen status and N₂O sources

Simulated oxygen content in the soil cores was close to that of the ambient air, with the modeled volumetric oxygen content ranging from 19.5 % to 20 % throughout the incubation period for all treatments (Table S7). Still, according to the model, denitrification-derived N₂O accounted for 76-100 % of the total emissions on average (Table S7).

In simulations, the 0-4 cm soil layer was treated as one uniform compartment, and this could have influenced model predictions. To investigate whether increasing the vertical resolution could improve the model performance, the soil profile was uniformly divided into five layers. The results showed that underestimations of high N₂O fluxes still existed after this change in the vertical representation of the model (Table S6). The simulated soil oxygen profiles were still predominantly aerobic for all treatments, as the single-layer model, but showed stratification over depth as depicted in Fig. 8. The oxygen level was the lowest at the beginning of incubation and then showed an increase over the period studied, mirroring the trends





of $CO_2$ flux. Despite the overall aerobic conditions in the soil, the large proportion of denitrification-derived $N_2O$ emissions was accompanied by the rapid growth of denitrifier biomass (data not shown).

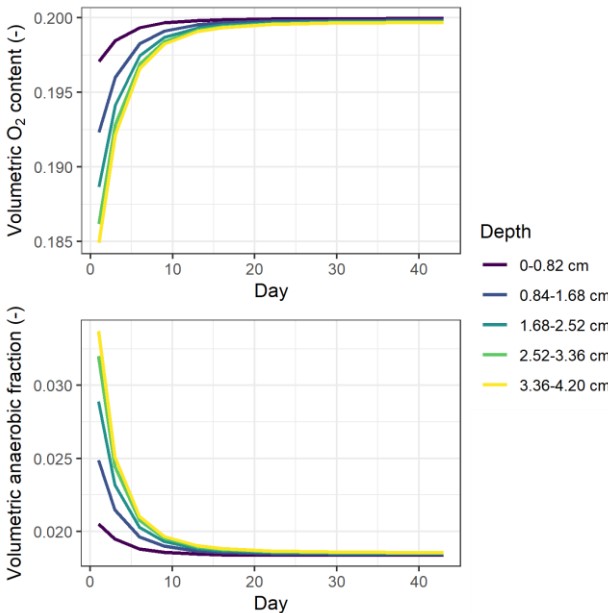

**Figure 8: Simulated $O_2$ content and volumetric anaerobic fraction by the multi-layer model for the treatment with the greatest soil**
**respiration rate (i.e. RC treatment with $NO_3^-$ addition at 60 % WFPS). The mean daily values from posterior runs were used here.**

### 3.2.5 Calibrated parameters

Although the number of parameters used for calibration had been reduced by Morris screening, more than half of the 26 calibrated parameters still exhibited random distributions within the predefined ranges (Fig. S4). The many potential inter-correlations may be the reason that these parameters could not be constrained to an unambiguous solution. But five
parameters were showing marked variability in their posterior distributions between treatments as depicted in Fig. 9, where the prior ranges of these parameters are indicated in the ordinate.

In most treatments, the parameter $d_{hrateNxOy}$ representing the N concentration for half rate in the denitrification process and also known as the Michaelis constant of the enzyme, was well constrained at the lower range of the parameter boundary within 50 mg N $L^{-1}$ in contrast to the mean of 250 mg N $L^{-1}$ in the prior range. A low $d_{hrateNxOy}$ relative to the physical
concentration of $NO_3^-$ resulted in a pronounced response of denitrifying bacteria activity to substrate availability (see Eq. (6.44) in Table S2). In some treatments which had no $NO_3^-$ addition, i.e. treatment 3, 7, 9, and 13, the parameter showed more diffused distribution and higher medians compared to other treatments. An enzyme with high $d_{hrateNxOy}$ relative to the concentration of substrate is not normally saturated with substrate and thus the rate of formation of product is substrate-limited.





In more than half of the treatments, the posterior distribution of $cn_m$, the microbial C/N ratio ($cn_m$) involved in calculations of mineralization and immobilization, was concentrated around 10 on average. However, for some $NO_3^-$ amended treatments which usually had higher $N_2O$ emission rates, i.e. treatments 4, 6, 10, 12, and 14, the distribution of calibrated microbial C/N ratio was not well constrained but similar to the prior distribution with medians up to 20.

The parameter representing the rate coefficient for the decay of the litter carbon pool, $k_{ll}$ generally showed higher values in

WW treatments than controls, and its range for the high nitrate RC treatment at 60% WFPS (treatment 4) was markedly higher than other treatments, indicating the faster decomposition of labile organic matter. Besides, the efficiency of NO-based denitrifier respiration, $d_{effNO}$, showed a low-range distribution for treatment 4. Low $d_{effNO}$ values induced a high respiration rate of denitrifiers for carrying out NO reduction (see Eq. (6.47) in Table S2). This treatment also exhibited a low-range distribution of the efficiency of SOM decomposition, $f_{e,ll}$, associated with a high fraction of $CO_2$ production.

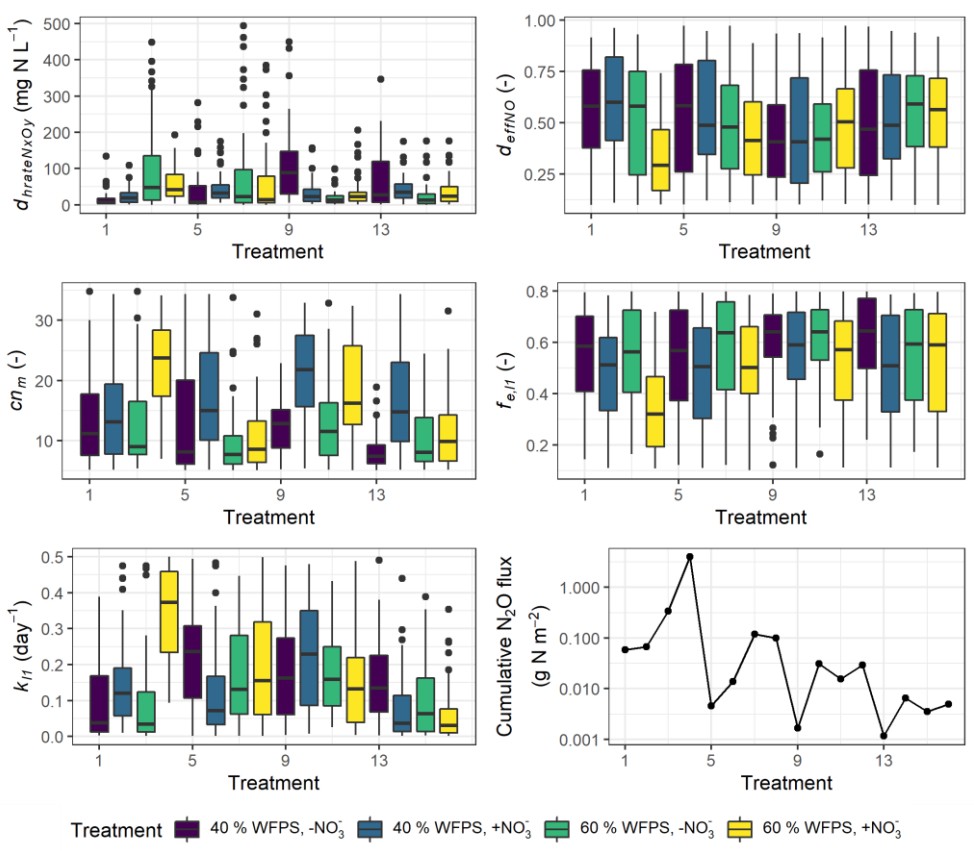


**Figure 9: Variability in five calibrated parameters among the 16 treatments. The boxplots show the 25 % and 75 % percentiles as the tops and bottoms of the boxes, and the medians as the bold lines. Treatment indices 1-4 represent treatments of mix RC, 5-8 for control RC, 9-12 for mix WW, and 13-16 for control WW, where treatment conditions are, in order: "40% WFPS, -NO₃⁻", "40% WFPS, +NO₃⁻", "60% WFPS, -NO₃⁻", and "60% WFPS, +NO₃⁻".**



## 4 Discussion

### 4.1 Sensitivity analysis

The sensitivity analyses, investigating the impact of parameter uncertainties on model predictions, revealed the importance of parameter interactions and connections between different processes in the model (Fig. 3a-f). Compared with $CO_2$ emissions and soil $NH_4^+$ content, $N_2O$ emissions were controlled by a larger number of parameters related to decomposition and denitrification processes and had generally lower SEEs. According to Fig. 3a, the sizes of humus and litter pools and their decomposition rates were critical in reducing the uncertainty of simulating $N_2O$ emissions. These results highlight the importance of reliable information on the initial state of the soil with respect to the composition and recalcitrance of organic matter pools where part of the organic matter within the soil cores could be fresh input from residues. The results confirm the findings of previous experimental and modeling studies showing the importance of substrate heterogeneity for decomposition and denitrification processes (Brilli et al., 2017; Eusterhues et al., 2003; Sierra et al., 2011). However, some other studies (Dungait et al., 2012; Schmidt et al., 2011) indicate that the chemical structure of organic molecules alone may not control their stability in soil, instead environmental and biological controls (e.g. accessibility of the SOM to decomposers, abiotic reactions, and desorption) predominate the SOM turnover especially in the longer term.

In our model simulations, C and N in crop residues were allocated to two labile pools, but the allocation ratio did not greatly influence $N_2O$ emissions, soil respiration, or mineral N. This could be related to the fact that the overall C/N ratio of crop residues was kept constant as the sizes of organic matter pools changed in the sensitivity analysis. The influence of crop residues on $N_2O$ emissions may be better reflected in other residue properties, e.g., the C/N ratio and solubility of individual substrates (Aulakh et al., 1991; Surey et al., 2020). Furthermore, it should be noted that addition of labile carbon from crop residues does not affect the decomposition of native soil organic matter in the model (i.e., no priming effect), as the decomposition of organic matter in labile and recalcitrant pools are calculated separately in CoupModel, similar as in other process-based models. The omission of a priming effect, the importance of which has been shown in field and laboratory studies (Kuzyakov, 2010), may cause models to underestimate the effects of crop residue composition on the turnover of soil C and N.

On the other hand, denitrifier growth parameters (e.g. $d_{effNO}$, $d_{growthNO3}$) showed considerable influence on the release of $N_2O$ in most treatments (Fig. 3a). Our results suggested that the influences of microbial activities on $N_2O$ emissions varied between different denitrification steps, and the denitrifier respiration for NO reduction showed a relatively larger and broader impact across treatments than other steps. The analysis of the two statistical measures $\sigma$ and $\mu$ suggested that, rather than a single factor driving the model to become more 'behavioral' in predicting $N_2O$ emissions, the collective effects of multiple parameters were more important, because one single parameter could exhibit various SEEs as other parameters changed, represented by high variability ($\sigma$) compared to the mean ($\mu$) in Fig. 3d. In calibrating complex models, several combinations of different parameter values might give the same goodness-of-fit between model outputs and measured variables, which is



defined as equifinality (Beven and Freer, 2001). The model sensitivity to such parameters is probably attenuated in the case of high-level equifinalities. Besides, the importance of parameter interaction structure associated with equifinality could hinder the constraint of parameters and hence the reduction of uncertainty in $N_2O$ simulations when limited measurement

data are available. For instance, the fraction of C mineralized to $CO_2$, characterized by $(1-f_{e,l1})$, and the decay rate of litter1 $(k_{l1})$, have a product interaction regarding the production of $CO_2$ (see Eq. (6.3) and Eq. (6.4) in Table S2). Also, the denitrifier growth rates (e.g. $d_{growthNO3}$) and the Michaelis constant characterized by $d_{hrateNxOy}$ influence the loss of N from anaerobic N pools by invoking microbial growth via a quotient interaction (see Eq. (6.41) and Eq. (6.44) in Table S2).

The parameters found to have the greatest impact on soil respiration and $NH_4^+$ content were associated with SOM

composition $(SOC_h)$ and decomposability $(k_{l1}, k_{l2}, f_{e,l1}, f_{e,l2})$, suggesting that model uncertainty for soil respiration and soil $NH_4^+$ could be greatly reduced if data for either SOM composition or decay rates were available. For simulating soil $NH_4^+$, information about microbial C/N ratio $(cn_m)$ and denitrifier growth parameters (e.g. $d_{growthNO3}$) is also important, because the availability of soil mineral N is closely associated with decomposition dynamics and its consumption by immobilization, nitrification, and denitrification (Lashermes et al., 2022). The influences of soil porosity and wilting point on $CO_2$ emissions

and soil $NH_4^+$ content were larger under, respectively, wet and dry conditions. The results can be explained by the fact that soil porosity and wilting point are key set points of the soil moisture response function controlling the upper and lower bounds of the function, which implies that the measurement of soil hydraulic properties could reduce model uncertainty under contrasting soil moisture levels.

### 4.2 Model performance and possible explanations for deviations

Overall, the performance of posterior models varied between estimated variables and treatments. The timing and magnitude of peak $N_2O$ emissions were more difficult to predict than those of $CO_2$ emissions even though parameters had been adjusted for individual treatments, and negative errors relative to observations were seen particularly when simulating high $N_2O$ emissions. Evaluation of model bias with respect to the slope $\beta_1$ in linear regression demonstrated a tendency across treatments to increasingly underestimate cumulative $N_2O$ flux as the observed flux increased. The problem of

underestimating high $N_2O$ fluxes by process-oriented models has been reported in previous studies. For example, Fang et al. (2015) showed that four different algorithms all underestimated the four highest cumulative $N_2O$ fluxes among eight N fertilizer treatments in an irrigated cornfield. Also, Gaillard et al. (2018) evaluated the simulated $N_2O$ flux from three process-oriented models (DNDC, DayCent, and EPIC) and reported an underestimation of 0.01-0.93 kg $N_2O$-N ha$^{-1}$ for every 1 kg of observed $N_2O$-N ha$^{-1}$ across models.

Residual analysis revealed that the model had a tendency to simultaneously underestimate $NO_3^-$ and overestimate $NH_4^+$ when $N_2O$ emission was underestimated, and this trend was even more pronounced when looking at control treatments only (Figs. S1 and S2). This suggests the nitrification rates may have been underestimated by the model and calls for revisiting the parameterization of the nitrification process. The simulated accumulation of $NH_4^+$ in the RC residue treatments, in contrast to





the transient NH$_4^+$ peaks observed (Fig. 5c), indicates that the modeled NH$_4^+$ release linked to decomposition was greater

than the NH$_4^+$ consumption by microbial immobilization and nitrification simulated. Nylinder et al. (2011) showed that a low nitrification rate simulated by the CoupModel was possibly the reason for the overestimation of the amount of soil NH$_4^+$ in a model of an organic cropping system. In our study, however, the weak and insignificant relationship between N$_2$O flux residuals and the residuals for mineral N indicates that N$_2$O underestimation at high flux ranges may be due to other factors.

Inaccurate estimation of proximal factors such as soil water content and temperature by the pedo-climatic subroutines has

been a main cause of errors in simulating C and N emissions in many process-based models (Brilli et al., 2017). In our study, the soil water content and temperature were assumed constant during incubation, but heterogeneity in the distribution of water could be a problem when initializing the soil environment in the model. Water retention capacity in the soil might be altered by the practice of adding crop residues. Lashermes et al. (2021) found that adding crop residues to soil increased the average WFPS of this layer from 60 % to 63 %. Kravchenko et al. (2017) found that specific gravimetric moisture of plant

residues in soil could vary in the range 60-220 %, and that residues were characterized by high moisture even at low soil water contents. Hence, the main effect of crop residues on the abiotic soil environment is probably not the marginal change in the average soil moisture content, but more likely the co-occurrence of elevated water content and labile C and N within the soil core. Residue fragments with high water retention capacity could represent microenvironments markedly different from those of the bulk soil and promote N$_2$O emissions (Kravchenko et al., 2017). Model results indicated that the simulated

O$_2$ content at 0-4 cm depth had only slight changes overall during incubation and was close to the saturation partial pressure in soil air owing to faster diffusion supply compared to soil respiration rates (Fig. 8). In the experiments, most of the oxygen consumption likely occurred in the microenvironment around residue debris. This is supported by observations of O$_2$ concentration in soil using O$_2$ microsensors (Markfoged et al., 2011) and planar optodes (Kravchenko et al., 2017) showing the aerated O$_2$ partial pressure in the soil matrix away from organic hotspots and steep gradients in O$_2$ between bulk soil and

hotspots of manure and residues, respectively. Nevertheless, simulations showed that denitrification was the major N$_2$O producing process in the experiment, accounting for 76-100 % of the total estimated N$_2$O emissions. Parkin (1987) found that a thin water film even as little as 20 μm, could be enough to deplete air and support denitrification at the surface of decaying litter, and it is thus possible that the observed high N$_2$O fluxes were produced via denitrification despite an overall high aeration status within the soil. Water absorption by residue fragments from the surrounding soil could create local

anoxic environments conducive to denitrification while also enabling the release of produced gases via drained pores (Kravchenko et al., 2018). In existing process-based models the heterogeneity in physical and biochemical processes caused by organic amendments is not included, which may limit the ability of these models to reflect the microscale anaerobiosis and SOC availability, and to predict peak N$_2$O emissions such as those observed in RC treatments (Fig. 5a). Some studies explored possibilities to incorporate spatial variability into denitrification models, although conceptual frameworks

considering heterogeneous environments for greenhouse gas emissions have only in recent years emerged and gained attention (Sihi et al., 2020). Using a stochastic modeling approach, Parkin (1987) found that the patchy dispersion pattern of



high denitrification microsites was a major factor influencing the overall rates of denitrification. Based on a parsimonious numerical model, Sihi et al. (2020) used probability distribution functions to represent soil microsite production and consumption of three greenhouse gases, which explained occasional observations of simultaneous $N_2O$ uptake (reduction)

and $CH_4$ uptake (oxidation) that were not typically captured by other models. We suggest that model development should improve on the description of microscale processes in soil, for example by parameterizing the distribution and extent of heterogeneity in, e.g., organic amendments and clay content, and by establishing the degree of anaerobiosis associated with hotspots and bulk soil separately.

The simultaneous underestimation of $N_2O$ and $NO_3^-$ could be linked to the incomplete description of nitrate supply in the

residue-amended 0-4 cm soil layer, which assumed there was no exchange with the lower 4-8 cm bulk soil layer. In a separate incubation experiment using the same soil type and several of the same treatments, Lashermes et al. (2021) found that adding RC residue to the 0-4 cm soil layer induced a decrease in the $NO_3^-$ content of the unamended 4-8 cm depth layer, indicating that the above, amended layer influenced the $NO_3^-$ dynamics in the bottom layer presumably caused by mass transfer between the two layers due to net consumption of $NO_3^-$ within the top layer during denitrification. In the current

model framework, solute transport is only simulated by convection (driven by water flow) and does not include diffusion driven by concentration gradients. The model was originally designed for field conditions, and at this spatial scale infiltration is the main mechanism for solute transport between compartments. However, in the short term after organic amendments, diffusive $NO_3^-$ supply from the bulk soil can be the most important source of electron acceptor for denitrification, as observed in earlier incubation studies (Nielsen et al., 1996; Petersen et al., 1996). The current solute transport process may

thus not be sufficient to properly simulate $N_2O$ production in microbial hotspots, especially under low flow rates or for short travel distances where diffusive flux becomes increasingly important (Flury and Gimmi, 2002). Microbial turnover could accelerate the recycling of N and increase substrate availability for nitrification and denitrification locally (De Bruijn et al., 2009), but in this and some other process-based models, microbial N is not connected to the mineral N pool or included in the calculation of total N budget, which could be another reason for model discrepancies in mineral N dynamics.

Only a few parameters showed distinct probability distribution patterns after calibration while others exhibited uniform distributions as the prior sampling. This result was in accordance with the second hypothesis, which should be related to the limited size of calibration data set in each treatment and the equifinalities between parameters. The differences in the posterior parameter distribution hold information about the characteristics of simulated C and N processes between treatments, although such variability may also reflect potential model limitations. For example, the well-constrained

microbial biomass C/N ratio ($cn_m$) within 10 in most treatments was consistent with observations that, on average, the C/N ratio of the soil microbial biomass varies between 6 and 10 at a global scale (Xu et al., 2013) and does not easily adapt in composition to litter quality (Spohn, 2015). Fungal cells typically have a C/N ratio ranging from 10 to 15 while bacteria range from 3.5 to 7 (Paul, 2007). In some treatments associated with extra $NO_3^-$ input and high $N_2O$ emissions, the microbial C/N ratios in accepted runs exhibited relatively high values, closer to the soil-residue mixture C/N ratio. According to Eq.



(6.7) and Eq. (6.8) in Table S2, a relatively high $cn_m$ could lead to a low level of humification (i.e. less labile C and N converted to recalcitrant matter) as well as intense N mineralization (i.e. more organic N in litter pools converted to $NH_4^+$). This could be associated with the underestimation of $NO_3^-$ availability discussed above, especially for the treatments amended with crop residues. Meanwhile, the relatively low values of estimated Michaelis constant $d_{hrateNxOy}$ suggested a high microbial affinity for soluble nitrogen oxides, accelerating microbial denitrification. In treatments without $NO_3^-$ addition, respiration via denitrification could be limited by the availability of electron acceptors through the respiratory chain, explaining an increase in the apparent Michaelis constant for N substrate reduction (Khalil et al., 2005). Including solute diffusion in the model may be able to change the posterior distributions of both parameters by better mimicking the mineral N supply. Compared to control treatments, the faster decay of organic matter ($k_{l1}$, $k_{l2}$) and higher $CO_2$ formation rate ($f_{e,l1}$, $f_{e,l2}$) in the crop residue treatments could reflect the need to mobilize N for use in nitrification and denitrification processes. Fast organic matter turnover in the residue-soil mixture was possibly caused by a high concentration of decomposer microorganisms associated with residue fragments. Additionally, in contrast to natural soils, human disturbance of the soil in the laboratory could stimulate indigenous microbial communities resulting in rapid biological phenomena (Calderon et al., 2001; Thiessen et al., 2013).

It should be noted that the model deviations for $N_2O$ flux were not caused by the spatial resolution of the vertical soil profile, which has been a problem in some studies (e.g. Xing et al., 2011), as the model performance concerning $N_2O$ prediction was not improved in the multi-layer model (Table S6) where the one-layer soil profile had been sub-divided into five layers for simulations. Deviations between modeled results and measured values are more likely to have resulted from limitations in the description of the N processes behind $N_2O$ emissions. For example, increasing the number of layers would not reflect the microscale processes associated with crop residue fragments and soil aggregates, nor would it address the missing description of solute diffusion between interfaces.

### 4.3 Treatment effects

We did not investigate how the model responded to the specific change of soil moisture and $NO_3^-$ level, but the results we obtained after calibrating the model against multiple treatments indicated the challenges in predicting $N_2O$ emissions under varying soil environmental conditions using a common model parameterization (Fig. 7). Similar cumulative $N_2O$ fluxes were simulated for treatments with the same $NO_3^-$ level regardless of the soil moisture level, which was different from observations. In the experiment, in RC treatments higher $N_2O$ fluxes were associated with the higher WFPS level (60 %) rather than with the higher $NO_3^-$ level, although there was a strong interaction between the two factors (Taghizadeh-Toosi et al., 2021). The problem to describe treatment effects of incubation studies by process-based models was discussed in a recent study by Grosz et al. (2021) who found that three $N_2O$ models (DNDC, CoupModel, and DeNi) responded to controlling factors in the same direction as measurements with frequencies from only 19 % to 67 %. Different from their study, in which no systematic calibration of model parameters was performed, the model deviations in our study, obtained by calibration of





multiple treatments, suggested that potential limitations in model assumptions or the description of mechanisms were more critical reasons for unsatisfactory model responses than parameterization. In CoupModel, while the denitrification subroutine is sensitive to changes in soil temperature, pH, mineral N, and SOC content, the soil moisture has indirect and average

effects on denitrification through decomposition, nitrification, and gas diffusion processes, but the effects of heterogeneity in the distribution of water and microbial activities are not represented. Therefore, soil moisture may have less effect on the $N_2O$ flux estimation in model applications than in real soil environments with heterogeneity in the distribution of C and N sources, and moisture. This is still one of the most challenging tasks facing soil biogeochemical models. On the other hand, our results showed a tendency to better predict treatments with higher $N_2O$ fluxes in the same group. This can be understood

from the characteristics of the calibration dataset and selection criteria. The high flux samples represented only a minor fraction of the total samples (i.e. 40 sampling points) in each group but were higher than the rest of them by orders of magnitude (Fig. 5a). The application of the ME criterion mainly constrained model deviations for the high fluxes in one data set, and less so for minor fluxes. It may be argued that this limitation could be improved by applying more stringent additional criteria such as $R^2$. However, this would reduce the acceptance rate or even refuse all posterior runs. Interestingly,

Vezzaro et al. (2012) obtained similar results in a GLUE context by using the Nash-Sutcliffe-based likelihood and stormwater measurements with large internal variability, and concluded that the choice of selection criteria should be based not only on its mathematical features but also by looking at the characteristics of the available data.

We also found that our capacity to evaluate model performance was limited by the data available for model estimation and calibration. Some model parameters were not assessed in the incubation experiment (e.g. soil/residue labile C content and

microbial biomass) and their values were either estimated or determined by calibration. The quality and temporal resolution in the measurement of controlling factors such as $NO_3^-$ and $NH_4^+$ were limited, and improving these aspects may reduce uncertainty in model prediction and facilitate model evaluation. By looking at the patterns of simulated $N_2O$ emissions and ancillary variables, we identified potential problems behind model principles, which should be investigated with experimental studies designed carefully for model use. Previous studies, including global sensitivity analyses (Metzger et al.,

2016; Wu et al., 2019) and model evaluations (Grosz et al., 2021), have specific suggestions to this end, such as improving measurement frequencies, evaluating sensitive input variables (e.g. decomposability of labile C), measuring more variables regarding N cycle (e.g. $N_2$, NO) and using state-of-the-art techniques (e.g. $^{15}N$ gas flux methods). We understand that collecting all data types discussed here is not always possible or practical, but encourage modelers to report more model outputs regarding N cycles even in the absence of observations, particularly the denitrification products, soil oxygen content,

and anaerobic fraction, which was not done very often in previous studies.

## 5 Conclusion

The current setup of CoupModel, when applied to results from an incubation study, indicated that parameters associated with the decomposability of SOM and denitrifier growth were important in regulating soil respiration and mineral N dynamics. A





high level of parameter interaction and equifinality issues existed regarding N$_2$O emissions, hindering the determination of
sensitivities and parameter constraints.

The parameters showing posterior distributions that differed from the prior distributions revealed specific modeled microbial
processes between treatments and may be used as references behind observations. For example, in the treatments without
NO$_3^-$ addition, the availability of N substrates to denitrifiers was limited according to the posterior distribution of Michaelis
constants. More intense SOM decomposition was simulated in residue treatments compared to controls.

The uncertainty analysis demonstrated a model bias towards underestimating high-range daily and cumulative N$_2$O fluxes,
which was associated with an inaccurate description of mineral N dynamics. Residual analysis indicated that nitrification
rate could be underestimated but did not sufficiently explain the model deviations. While the simulated soil respiration
response to soil moisture was generally in line with the direction of measurement, the modeled N$_2$O emissions were not as
sensitive to the WFPS as the measured data, probably because of the indirect effect of soil moisture response function on the
denitrification process. Discussing potential limitations in model principles related to the prediction bias, we described
several suggestions for model improvement including the use of new parameters and equations to represent microscale
heterogeneity, and a re-examination of the effects of soil moisture on denitrification processes.

Generally, we conclude that modeling N$_2$O emissions in controlled experiments is useful to identify the need for prior
knowledge in both basic (e.g. decomposability of SOM) and elaborate (e.g. denitrifier growth) aspects of the process-based
model for reducing the uncertainty of N$_2$O flux estimates. Moreover, we identified a potential model bias and discussed
future steps that may be required to assess its sources. We believe there is a need to modify model equations and revisit basic
model assumptions with high-quality measurement data sets that enable more intensive model evaluations and comparisons.

**Acknowledgments**

This study was financially supported by Independent Research Fund Denmark (DFF) (Project acronym: modelN2O). Dr.
Arezoo Taghizadeh-Toosi is acknowledged for having provided the experimental data. We further thank Iris Vogeler Cronin
for providing insights on the model results. Zhang W. acknowledged grant from the Swedish Research Council VR 2020-
05338.





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
