# Peer review of "Modeling nitrous oxide emissions from agricultural soil incubation experiments using CoupModel"

_Biogeosciences, 2022_

## Author Response (AR1)

We thank the reviewers for taking the time to review our manuscript and provide constructive comments. Their efforts towards improving our manuscript were much appreciated. We have made significant modifications reflecting those comments, which are outlined point by point below and relevant changes are marked in bold.

**Reply on RC1**

**RC comment:** The manuscript is well written and provides an interesting review of challenges to biogeochemical modeling of $N_2O$ fluxes. It presents an incubation experiment paired with modeling to better resolve drivers of error in $N_2O$ modeling. However, the most interesting discovery from that effort, which is the contribution of biases in $NO_3^-$ and $NH_4^+$ towards $N_2O$ flux biases, is touched on fairly superficially and should be delved into in much more detail.

**Response:** We agree with the reviewer that regarding contributions of biases in mineral N and associated $N_2O$ fluxes, we should investigate it in more detail. In the residual analysis in Figs. S1-S2, we found that the residual errors in $NO_3^-$ and $NH_4^+$ were only weakly correlated with the residual errors in $N_2O$. In lines 534-535 of the revised manuscript, it was stated: "the relationships between $N_2O$ flux residuals and the residuals for mineral N were weak and not significant, indicating that the $N_2O$ underestimation at high flux ranges may also be due to other factors". We are also aware that in cases where the calibrated model failed to capture the magnitude of pulse fluxes in specific treatments, the variability in posterior model ensembles (Fig. 5a) indicated that there may be large uncertainties in parametrization.

In the experiment, there was a consistent increase in $NH_4^+$-N content by day 1 after red clover residue application, whereas in the model, soil organic matter decomposition followed first-order kinetics which tended to cause a gradual increase of $NH_4^+$ over the simulation rather than the rapid increase observed in the early stage. According to Lashermes et al. (2022) the red clover residues contained water-soluble N (WSN) corresponding to c. 1 g N m$^{-2}$ which can account for the mineral N released. Cutting and mixing residues into the soil in the incubation experiment probably accelerated release and mineralization of N from the WSN pool compared to a realistic field situation (Angers and Recous, 1997), and this challenges a model that is calibrated to describe decomposition in natural environments. **We added some model sensitivity tests to demonstrate how the magnitude of peak flux is sensitive to initial mineral N content and the effect of dynamic parameters (see lines 637-662 and Figs. S7 and S8) attempting to improve modeling practices.**

**RC comment:** Subsequently, the paper discusses many potential drivers of model error and challenges in experimentation to better identify and address contributions to this error. However, the study conducted does not help address these shortcomings much at all. Hence, in my opinion this article is of limited value as an original research paper and is in fact a mix of limited original research and interesting review. I urge the researchers to push towards work to unravel these meaningful issues they coherently discuss here.

**Response:** We agree with the reviewer that we should address these shortcomings of the model and discuss potential attributors, besides, we want to emphasize the value of our article as an original research paper for the following reasons:

Process models are primarily used for field-scale simulations where the discussion of model deviations with respect to N$_2$O estimation often refers to inaccurate pedo-climate subroutines (Brilli et al., 2017). The present study focused on the role of reactive C and N for N$_2$O emissions, and used simulation of targeted experiments to identify key drivers. Simulating C and N dynamics in a short-term laboratory study may be considered to zoom in on a single field operation, in this case the incorporation of crop residues by standard tillage operations. Grosz et al. (2021) in a recent paper in Biogeosciences applied three process models, including CoupModel, to results from an incubation study, but without model calibrations. In contrast, the present study examined model parameters with respect to soil physical properties, but also decomposition, nitrification and denitrification, to learn about sensitivities and limitations.

The unamended soil treatments (controls) were well described by the model, indicating that the repacked and preincubated soil was a suitable representation of normal soil conditions. In contrast, C and N dynamics in residue-amended soil, especially at the high moisture level (Fig. 4a), were difficult to describe with the model, possibly because of the mixing of residues and soil accelerating C and N turnover.

The simulation of predominantly aerobic conditions on the basis of soil respiration and oxygen diffusion, despite significant N$_2$O emissions, has not been reported before. Also, parameters related to denitrification and other processes ranked high in the sensitivity analyses, while there was a weak response of estimated N$_2$O fluxes to soil moisture. This could be related to heterogeneous moisture distribution in the soil-residue mixture in the experiment, indicating the presence and importance of organic hotspots.

**We revised the Introduction to better explain the motivation and focus of the study (see lines 93-96/ lines 108-110), and we added a new section to the Discussion on these modeling outcomes and need for better experimental design (see lines 637-678).**

Other comments:
**RC comment:** How were the parameter ranges derived? It's insufficient to just describe them as "with realistic ranges" or according to model defaults. The ranges are important to model sensitivity and calibration equifinality issues.

**Response: The ranges of input parameters were derived around the mean values of measurements and estimations, and details were added in Table S4, and justified in the text (lines 195-210)**. For parameters supported by measurements, i.e. soil porosity and pH, the ranges were within 25% of the mean values to represent realistic micro-scale variations in the laboratory setup. The residue porosity was estimated with a bulk density of 0.18 g/cm$^3$ and a dry density of 1.3 g/cm$^3$ similar to the reported bulk densities and porosities of crop residues in experimental studies with a wider range of uncertainty due to compressibility. Soil porosity and residue porosity were used to calculate the soil-residue mixture bulk density and porosity which was beyond the CoupModel framework, and this description was added in the revised Supplement. The range of organic pool fractions were considered from the literature values for cultivated soil as described in the text**,** and for crop residues the range was bound by the estimated fractions of two crop residues**.**

Regarding process parameters, we looked through most relevant calibration studies using CoupModel and other process models, and adopted ranges defined in the model on the basis of previous applications as shown at the bottom of Table S4. The ranges of remaining model-specific parameters

have not been reported in the existing literature, including those involved in nitrification and denitrification, we adopted the default ranges set by the model without better prior information.

**RC comment:** Too much is shown in the figure 5 subplots for interpretation. This data needs to represented in a better manner.

**Response:** Agree. **We split Fig. 5 into two separate Figures by WFPS level. One Figure with treatments at 60% WFPS (with the most interesting gas emission dynamics) was kept in the manuscript (see new Fig. 5), while a Figure with treatments at 40% WFPS was be included in the supplement (see new Fig. S6).**

**RC comment:** In table 1, why is the rRMSE so much different between the single treatment and multi-treatment for $NH_4^+$?

**Response:** For $NH_4^+$ in the multi-treatment calibration, high rRMSE was almost exclusively caused by larger biases between measured data and modeled data for red clover treatments compared to those in the single-treatment calibrations. This has been depicted in Figs. 6c and 7c, please note that a log-scale is used. The statistics rRMSE and RMSE are more sensitive to outliers compared to mean error (ME) and they would be reduced to 78% and 0.01, respectively, after removing the four treatments (green points in Fig. 7c). **In the revised paper, we added relevant sentences in section 3.2.3 (lines 396-399).**

**RC comment:** You describe a pattern of better model fit as the simulations progress with time. This sounds like a model initialization issue. Did you make any attempts to spin-up the model?

**Response:** We did not spin-up the model but took the initial carbon pool sizes from literature values. Spin-up is intended to make the soil pools reach an equilibrium between carbon input and decomposition, which is often the assumed state in long-term studies of ecosystems (Hashimoto et al., 2011). In our study, we assumed that the pre-incubated bulk soil was already close to a steady state at the start of incubations, in accordance with the C and N dynamics in control treatments, and that the changes observed during incubation were mainly caused by decomposition of residues introduced at time zero. **A statement was included to clarify the initialization step (lines 163-166).**

**RC comment:** Do you have ideas of what caused the second flux peak? Was it the residue decomposition? Something else?

**Response:** The observed secondary increase in $CO_2$ fluxes from treatments with wheat straw was most pronounced in treatments with elevated soil $NO_3^-$ (see Taghizadeh-Toosi et al., 2020; Fig. 1), and presumably it reflected growth of heterotrophic microorganisms, which could be enhanced by assimilatory $NO_3^-$ reduction. Without specific evidence, however, we prefer not to speculate on this.

**RC comment:** Isn't seeing ranges of calibrated parameters oscilating heavily across treatments a sign that the calibration is largely fitting noise?

**Response:** We understand the "noise" as measurement error. We used mean error (ME) as the main measure of calibration, aiming to have a ME close to zero by fitting daily measurements. Considering measurement errors often are normally distributed around zero, and thus their influence on parameter fitting may have been limited if we rule out serious experimental systematic errors. The limitation of the measurement dataset is that we only had mean values of recalculated measurements to compare with

model simulations, and therefore any measurement error was included in parameter ranges. Future experiments should have explicit measurements in separate layers with error bars for model use. **We added further relevant statements about measurement dataset limitations to the Discussion (see lines 663-678).**

References:

Angers, D. A. and Recous, S.: Decomposition of wheat straw and rye residues as affected by particle size, Plant Soil 1997 1892, 189, 197–203, doi:10.1023/A:1004207219678, 1997.

Brilli, L., Bechini, L., Bindi, M., Carozzi, M., Cavalli, D., Conant, R., Dorich, C. D., Doro, L., Ehrhardt, F., Farina, R., Ferrise, R., Fitton, N., Francaviglia, R., Grace, P., Iocola, I., Klumpp, K., Léonard, J., Martin, R., Massad, R. S., Recous, S., Seddaiu, G., Sharp, J., Smith, P., Smith, W. N., Soussana, J.-F. and Bellocchi, G.: Review and analysis of strengths and weaknesses of agro-ecosystem models for simulating C and N fluxes, Sci. Total Environ., 598, 445–470, doi:10.1016/j.scitotenv.2017.03.208, 2017.

Lashermes, G., Recous, S., Alavoine, G., Janz, B., Butterbach-Bahl, K., Ernfors, M. and Laville, P.: $N_2O$ emissions from decomposing crop residues are strongly linked to their initial soluble fraction and early C mineralization, Sci. Total Environ., 806, 150883, doi:10.1016/j.scitotenv.2021.150883, 2022.

Hashimoto, S., Wattenbach, M. and Smith, P.: A new scheme for initializing process-based ecosystem models by scaling soil carbon pools, Ecol. Modell., 222, 3598–3602, doi:10.1016/j.ecolmodel.2011.08.011, 2011.

Taghizadeh-Toosi, A., Janz, B., Labouriau, R., Olesen, J. E., Butterbach-Bahl, K. and Petersen, S. O.: Nitrous oxide emissions from red clover and winter wheat residues depend on interacting effects of distribution, soil N availability and moisture level, Plant Soil, 466, 121–138, doi:10.1007/s11104-021-05030-8, 2021.

**Reply on RC2**

**RC comment:** Zhang et al. present a study where results from a short term incubation study was used to assess the ability of a process oriented model to simulate $N_2O$ emissions from soil with the addition of different crop residues and nitrate levels. The paper is generally well written and the topic within the scope of Biogeosciences and presents an interesting discussion and review about challenges of accurately simulating soil $N_2O$ fluxes. Therefore, I think, that the manuscript should be valuable for other researchers trying to model $N_2O$ emissions from soils and could be potentially published in Biogeosciences.

**Response:** Thank you for acknowledging our efforts to diagnose the behavior and test the performance of a process model in the realistic scenario for agroecosystems where C and N availabilities suddenly change as observed in a laboratory setup with soil-residue mixtures with corresponding controls.

**RC comment:** However, I agree with the comment by Lorenzo Brilli, that the approach is not really novel and that there is the need to focus more on solutions rather than discussing the limitations.

**Response:** While the methods used in this study are not novel, there are very few papers available to implement a global sensitivity and uncertainty analysis based on a complex framework which includes the fundamental C and N processes in the soil associated with classical formulas that describe soil physics (e.g., soil water, heat coupled transport, gas diffusion etc.).

With a consistent modeling approach, we evaluated the performance of the model in simulating $N_2O$ emissions from soil mixed with two contrasting residues, and the results give some directions for future improvements of simulation experiments that could be useful to modelers and for experimental design. It is well-known that heterogeneity at mm-scale in the distribution of moisture as well as labile C and N is critical for $N_2O$ emissions (Kravchenko et al., 2017; Parkin, 1987), and the results of the present study suggest that residue particles were an important source of $N_2O$. The effects of concurrent C and N transformations in organic hotspots can lead to intensive biological activity and process models may require new solutions to describe this. We do believe this model evaluation work has value, even if it was beyond our research goals to investigate specific solutions here. **Possible solutions and relevant future work on the heterogeneity issue was briefly discussed lines 560-569 and lines 617-621. These implications of the study should have been stressed more, and we included new modeling tests in the sensitivity of dynamic parameters and initial mineral N content in the Discussion about possible strategies in improving modeling practices under current framework (see lines 637-662 and Figs. S7 and S8).**

**RC comment:** In addition, I am not really sure to what extent results from a short term incubation, with sieved and repacked soil cores and limited measurements can be used to calibrate and quantify the uncertainty of a process based model used for simulating N cycling and $N_2O$ emissions under field conditions. The conditions used in the incubation (sieved, repacked cores, constant temperature and soil moisture) are not typically found in the field and highest $N_2O$ emissions are often associated with wetting and drying cycles.

**Response:** Thanks for sharing this fair concern which has highlighted the need to discuss the connection between incubation and field conditions. As mentioned also in the response to RC1, the short-term experiment used here for model evaluation represented the period after residue incorporation with an

instantaneous input of labile C and N to the soil, the difference between a long-term and a short-term study being whether the incorporation happened at some point during the simulation or initially.

Both field and incubation studies share the same biotic processes, i.e., decomposition, nitrification and denitrification, and in the past model equations have been derived from both types of studies (e.g. DNDC Scientific Basis and Processes, 2017). While models are intended to be used in the field, targeted laboratory experiments are often used to test submodules under controlled conditions (e.g. Grosz et al., 2021; Xing et al., 2011).

The experiment with sieved and repacked soil in cylinders with a change in soil moisture content and temperature compared to previous storage did not show evidence for major disturbances of soil C and N turnover when preincubated for a week under the new conditions before the experiment. In contrast, the residue amendment stimulated biological activity including $N_2O$ emissions, and this would also be the case under field conditions. We acknowledge that the time course of C and N turnover was probably faster in these incubations than in the field due to the mixing of residues and soil, and this can help explain why in some cases simulations by the model, calibrated to describe field observations, could not capture the dynamics.

We agree that high $N_2O$ emissions are often associated with wetting and drying cycles, but many studies (e.g. Duan et al., 2017) have also shown that decomposer activity can sustain anaerobic processes and $N_2O$ production in well-drained soil. Currently the importance of experimental conditions, and the potential for accelerated C and N turnover in organic hotspots is not mentioned until the discussion. **We expanded the Introduction to mention this important context for the modeling of residue decomposition (see lines 49-56).**

**RC comment:** Moreover, sieving the soil will result in the destruction of soil aggregates and lead to increased SOM mineralization. I think that theses points need to be better highlighted in the paper and their implications for modelling $N_2O$ emissions under field conditions discussed.

**Response:** We tend to disagree that sieving to <6 mm is a major disturbance. Petersen and Klug (1994) did not find a significant difference in $CO_2$ evolution when incubating a similar arable soil after sieving <2 or <4 mm. Also, there was no stimulation of $CO_2$ emissions from the sieved control soil for WW treatments (a minor increase in control soil for red clover treatments was associated with high moisture). Therefore, the decomposition was most likely dominated by the fresh residues.

**RC comment:** LN 25 ff: "For the development of process-based models, we suggest there is a need to address soil heterogeneity, and to revisit current subroutines of moisture response functions." Soil heterogeneity was very much reduced in this experimental set up by sieving and repacking the soil. Can you comment what this implies for field measurements?

**Response:** For the soil environment investigated in the incubation study, the heterogeneity of importance to $N_2O$ emissions was mainly that associated with residue fragments, since this is where labile C and N were concentrated. In comparison, bulk soil heterogeneity, or the reduction of heterogeneity caused by sieving, was probably much less important. In the experiment, heterogeneity in the distribution of residues was reduced compared to field conditions, since residues were cut and mixed into the soil. However, the residue-soil contact was increased compared to field conditions, and in fact this may have increased the effects of heterogeneity. Although mixing should make the distribution

of C and N more similar to the distribution assumed in the model, it may have accelerated residue decomposition and the associated $N_2O$ emission compared to the field situation.

**RC comment:** Ln 108, to what size was the soil sieved?

**Response:** The soil was sieved to <6 mm (Taghizadeh-Toosi et al., 2021). **We added the information in the Method section (see line 116).**

References:
Duan, Y. F., Kong, X. W., Schramm, A., Labouriau, R., Eriksen, J. and Petersen, S. O.: Microbial N transformations and $N_2O$ emission after simulated grassland cultivation: Effects of the nitrification inhibitor 3,4-dimethylpyrazole phosphate (DMPP), Appl. Environ. Microbiol., 83, doi:10.1128/AEM.02019-16/FORMAT/EPUB, 2017.

Kravchenko, A. N., Toosi, E. R., Guber, A. K., Ostrom, N. E., Yu, J., Azeem, K., Rivers, M. L. and Robertson, G. P.: Hotspots of soil $N_2O$ emission enhanced through water absorption by plant residue, Nat. Geosci., 10, 496–500, doi:10.1038/ngeo2963, 2017.

Parkin, T. B.: Soil microsites as a source of denitrification variability, Soil Sci. Soc. Am. J., 51, 1194–1199, doi:10.2136/sssaj1987.03615995005100050019x, 1987.

Petersen, S. O. and Klug, M. J.: Effects of sieving, storage, and incubation temperature on the phospholipid fatty acid profile of a soil microbial community, Appl. Environ. Microbiol., 60, 2421–2430, doi:10.1128/aem.60.7.2421-2430.1994, 1994.

Taghizadeh-Toosi, A., Janz, B., Labouriau, R., Olesen, J. E., Butterbach-Bahl, K. and Petersen, S. O.: Nitrous oxide emissions from red clover and winter wheat residues depend on interacting effects of distribution, soil N availability and moisture level, Plant Soil, 466, 121–138, doi:10.1007/s11104-021-05030-8, 2021.

Xing, H., Wang, E., Smith, C. J., Rolston, D., and Yu, Q.: Modelling nitrous oxide and carbon dioxide emission from soil in an incubation experiment. Geoderma, 167, 328-339, doi:10.1016/j.geoderma.2011.07.003, 2011.

**Reply on CC1**

**CC comment:** The manuscript provide results from a short-term (43-day) factorial incubation experiment to investigate the ability of a process-oriented model (CoupModel) to estimate $N_2O$ and carbon fluxes, and soil mineral nitrogen (N) dynamics. The manuscript is well written, it fluently flows and the whole structure is coherent with the adopted approach. To my opinion, all three objectives indicated by authors at the end of the introduction were satisfying investigated. This would make the paper suitable to be published.

**Response:** Thank you for appreciating our effort in conducting the modeling study. It was a lot of work and we did our best to diagnose model behavior and test the performance.

**CC comment:** However, many similar works were developed and published through years, reporting similar issues and conclusions. This make the novelty of the paper very poor, despite the large work done. To overcome this huge limitation, I suggest authors to be more proactive at presenting solutions on how to solve the detected issues under current modelling limitations. Within the text, in fact, only general suggestions to cope with these issues are reported (i.e., revisit basic model assumptions and equations, increasing high-quality measurement data, etc.), but none proper solution (new equations to implement and their description, description of further steps in soil incubation experiments, previous chemical analysis to do, etc.) and related changes in final results were reported. I understand that this is not the primarily objective declared within the paper, but since an exponential number of modelling works were published in the last 30 years, a step forward to indicate how to overcome these limitations would be done.

**Response:** In the responses to RC1 and RC2 we have tried to argue what makes this study unique. The incubation experiment providing the data used for modeling may be considered to zoom in a common field operation with residue recycling that may result in a period with high $N_2O$ emissions in the field depending on residue quality. We have challenged the model to simulate results from short-term incubations to learn about sensitivities and limitations. The simulation showed predominantly aerobic conditions, but still $N_2O$ emissions were substantial, which supports experimental studies demonstrating the importance of mm-scale hotspots for $N_2O$ emissions. Both the high $O_2$ availability predicted by the model and the observed $N_2O$ emissions are consistent with the measured $O_2$ distribution and coupled nitrification-denitrification around manure hotspots (e.g., Petersen et al., 1996) and red clover hotspots (Kong et al., 2017). Also, weak responses of estimated $N_2O$ fluxes to soil moisture could be related to heterogeneous moisture distribution in the real soil-residue mixture and the implicit role of soil moisture response function in denitrification module as discussed in section 4.2 and 4.3. As discussed earlier, the gap between model simulations and the most intense $N_2O$ emissions could be partly due to experimental design, and partly due to constraints on parameters associated with denitrification. We agree with the Lorenzo Brilli that further statements on how to overcome limitations could be done. **We included a new section in the Discussion about improving modeling practices and better experimental design (lines 637-678 and Figs. S7 and S8).**

References:
Kong, X., Duan, Y., Schramm, A., Eriksen, J., Holmstrup, M., Larsen, T., Bol, R. and Petersen, S. O.: Mitigating $N_2O$ emissions from clover residues by 3,4-dimethylpyrazole phosphate (DMPP) without

adverse effects on the earthworm Lumbricus terrestris, Soil Biol. Biochem., 104, 95–107, doi:10.1016/j.soilbio.2016.10.012, 2017.

Petersen, S. O., Nielsen, T. H., Frostegård, Å. and Olesen, T.: $O_2$ uptake, C metabolism and denitrification associated with manure hot-spots, Soil Biol. Biochem., 28, 341–349, doi:10.1016/0038-0717(95)00150-6, 1996.